# CrossMed: A Multimodal Cross-Task Benchmark for Compositional Generalization in Medical Imaging

Pooja Singh*, Siddhant Ujjain*, Tapan Kumar Gandhi, and Sandeep Kumar

*Indian Institute of Technology Delhi, New Delhi, India*

{eez228470, eez228484, tgandhi, ksandeep}@iitd.ac.in

*Abstract*—Recent advances in multimodal large language models (MLLMs) has enabled unified processing of visual and textual inputs, with promising implications for general-purpose medical AI. However, their ability to generalize compositionally across unseen combinations of imaging modality, anatomy, and task type remains underexplored. We introduce CrossMed, a benchmark designed to evaluate compositional generalization (CG) in medical MLLMs using a structured Modality–Anatomy–Task (MAT) schema. CrossMed reformulates four public datasets, CheXpert (X-ray classification), SIIM-ACR (X-ray segmentation), BraTS 2020 (MRI classification and segmentation), and MosMedData (CT classification) into a unified visual question answering (VQA) format, resulting in 20,200 multi-choice QA instances. We evaluate two open-source MLLMs, LLaVA-Vicuna-7B and Qwen2-VL-7B, on both *Related* and *Unrelated* MAT splits, as well as a zero-overlap setting where test triplets share no Modality, Anatomy, or Task with training data. Models trained on Related splits achieve 83.2% classification accuracy and 0.75 segmentation cIoU, while performance drops significantly under Unrelated and zero-overlap conditions, validating the benchmark's difficulty. Furthermore, we show cross-task transfer where segmentation performance improves by +7% cIoU even when trained using classification-only data. Traditional models (ResNet-50, U-Net) benefit modestly, confirming MAT's broad utility, while MLLMs uniquely excel at CG. CrossMed provides a rigorous testbed for evaluating zero-shot, cross-task, and modality-agnostic generalization in medical vision-language models.

*Index Terms*—Multimodal large language models (MLLMs), medical vision language understanding, compositional generalization, visual question answering (VQA), MAT triplet schema.

## I. INTRODUCTION

Medical imaging plays an essential role in modern healthcare, facilitating diagnosis, monitoring, and treatment of various diseases. Recent advances in deep learning have substantially improved performance in medical tasks, including classification [1], detection [2], and segmentation [3]. However, these approaches typically depend heavily on large-scale annotated datasets and task-specific architectures, limiting their ability to scale and generalize across diverse clinical tasks and imaging modalities. Traditional single-task models, although effective within narrow scopes, struggle with knowledge transfer, especially in scenarios involving rare or

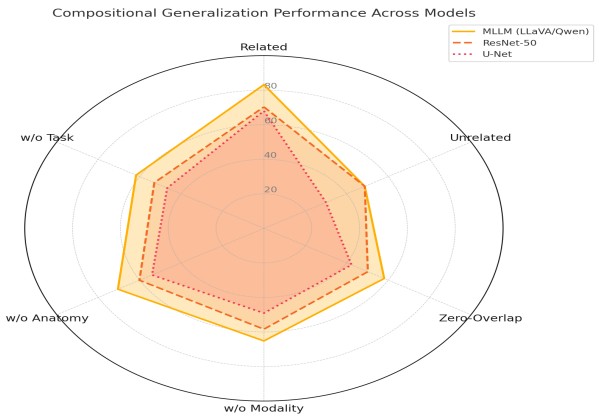

Fig. 1. Classification accuracy across compositional generalization conditions: *Related*, *Unrelated*, with holding individual MAT factors (w/o Modality, w/o Anatomy, w/o Task), and *All Data* multi-task upper bound.

emerging medical conditions [4]. While multi-task learning approaches have demonstrated promise for enhancing flexibility and generalization, the fundamental mechanisms enabling this cross-task performance remain inadequately understood [5].

Recently, multimodal large language models (MLLMs), which combine visual and textual reasoning capabilities, have emerged as powerful tools showing promising results across a range of biomedical applications [6], [7]. Nonetheless, a clear understanding of how these models generalize across heterogeneous medical datasets remains elusive. We hypothesize that compositional generalization (CG), the ability of models to recombine learned elements to address novel tasks and scenarios, plays a crucial role [8]. For example, in Fig. 2, a model trained on "white rabbit" and "black piglet" should be able to infer "black rabbit" even if it has never seen this combination before. In medical imaging, each sample can be broken down into a Modality–Anatomy–Task (MAT) triplet, such as (*X-ray*, *Chest*, *Classification*) or (*MRI*, *Brain*, *Segmentation*). CG suggests that a model trained on certain combinations should be able to generalize to unobserved ones, like (*MRI*, *Chest*, *Classification*).

To systematically explore this hypothesis, we introduce CrossMed, a comprehensive benchmark explicitly designed to evaluate CG in medical vision-language models through

---

*These authors contributed equally to this work

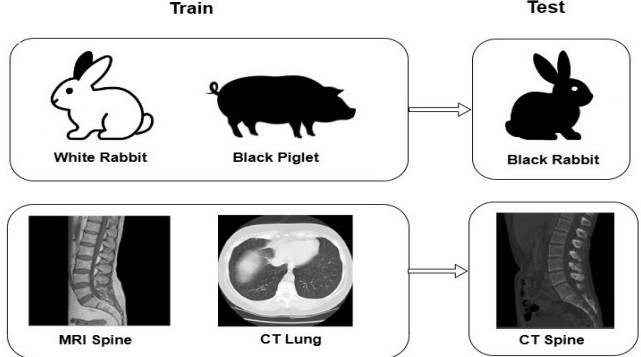

Fig. 2. Illustration of compositional generalization under Related vs. Unrelated training splits. Top: A model trained on White Rabbit and Black Piglet images generalizes to an unseen Black Rabbit by recombining color and object attributes. Bottom: Training on MRI Spine and CT Lung enables correct interpretation of a novel CT Spine image, demonstrating generalization through shared modality–anatomy factors.

a standardized visual question answering (VQA) paradigm. CrossMed contains 20,200 MAT-labeled examples spanning multiple imaging modalities (X-ray, MRI, CT), rigorously organized into Related and Unrelated splits to precisely isolate compositional effects. Further, we implement a strict zero-overlap evaluation to robustly assess true zero-shot generalization. Our experiments, conducted with two state-of-the-art MLLMs (LLaVA-Vicuna-7B and Qwen2-VL-7B), reveal significant performance improvements under compositional training conditions, confirming the effectiveness and scalability of the CrossMed benchmark.

**Key Contributions:**

- A unified, multi-task VQA benchmark encompassing a diverse set of medical imaging modalities, anatomies, and tasks.
- A rigorous zero-overlap evaluation protocol designed to provide a reliable measure of compositional generalization.
- Empirical evidence demonstrating robust generalization capabilities of MLLMs to novel MAT combinations.
- CrossMed supports both task-level and modality-level transfer, and enables consistent evaluation across architectures via a unified format.

## II. RELATED WORK

Multimodal large language models (MLLMs) have catalyzed major advances in visual-language reasoning, enabling capabilities such as visual question answering (VQA), captioning, and instruction-following [9]–[11]. These models typically combine powerful pre-trained language backbones with vision encoders, resulting in remarkable zero-shot and few-shot generalization in natural image domains. However, their extension to the medical domain remains nascent [12], with most work limited to narrow tasks like report generation or classification in specific imaging modalities. Traditional medical AI systems, largely based on convolutional or transformer architectures, are

effective but siloed often built for task-specific deployments such as disease classification or lesion segmentation [13], [14]. While some success has been achieved through transfer learning and self-supervision, these models often fail to generalize across tasks or modalities due to the lack of shared structural representations [15], [16]. Recent adaptations of MLLMs to medicine have explored integrating language modeling with visual features for structured outputs. HuatuoGPT-Vision [7], PMC-VQA [17], and MedFlamingo [18] represent promising directions by aligning biomedical images with instruction-tuned language models. However, these approaches are primarily evaluated on narrow datasets and do not systematically examine generalization under novel task configurations. The concept of compositional generalization (CG), a model's ability to recombine familiar components to solve unfamiliar tasks has gained traction in both NLP and computer vision. In the medical domain, Med-MAT [19] marks a notable contribution by curating a suite of 106 public datasets and constructing Modality–Anatomy–Task (MAT) triplets for classification and detection tasks. It evaluates zero-shot recombination performance and provides a strong baseline for CG in medical MLLMs. However, Med-MAT does not integrate tasks into a single inference format, nor does it address segmentation or provide theoretical bounds on CG.

In contrast, our CrossMed benchmark unifies medical imaging tasks spanning classification and segmentation into a single VQA-based interface. Unlike Med-MAT, CrossMed provides a principled factorization of CG using MAT triplets and introduces rigorous Related, Unrelated, and zero-overlap evaluation splits. We extend the scope of generalization to include continuous mask decoding and task transfer (e.g., classification-only training with segmentation evaluation). CrossMed is also empirically validated on multiple MLLMs and compared with traditional baselines (e.g., ResNet, U-Net), underscoring its clinical and methodological robustness. Our work bridges a critical gap by offering both a unified evaluation protocol and empirical analysis of CG effects across diverse architectures and imaging modalities, moving toward the goal of generalist medical AI systems.

## III. DATASET CONSTRUCTION

To evaluate compositional generalization in medical MLLMs, we construct the CrossMed benchmark, comprising five MAT configurations across multiple public datasets (Table I). Each configuration represents a clinically relevant combination, enabling fine-grained control over compositional overlap.

TABLE I
MAT TRIPLETS USED IN CONSTRUCTING THE CROSSMED BENCHMARK.

| Modality | Anatomy | Task | Source Dataset |
|----------|---------|------|----------------|
| X-ray | Chest | Classification | CheXpert |
| X-ray | Chest | Segmentation | SIIM-ACR (Pneumothorax) |
| MRI | Brain | Classification | BraTS 2020 (Glioma grade) |
| MRI | Brain | Segmentation | BraTS 2020 (Tumor masks) |
| CT | Lung | Classification | MosMedData (COVID-CT) |

- **(X-ray, Chest, Classification)**: From CheXpert [1], each image is paired with a binary label (e.g., "Is pneumonia present?") in a VQA format.
- **(X-ray, Chest, Segmentation)**: From SIIM-ACR [20], pneumothorax regions are segmented, and models select the correct binary mask among four options.
- **(MRI, Brain, Classification)**: Based on BraTS 2020 [3], glioma subtype grading is posed as a multi-class classification question.
- **(MRI, Brain, Segmentation)**: Tumor masks from BraTS 2020 are used in a four-choice segmentation task.
- **(CT, Lung, Classification):** From MosMedData [21], CT scans are labeled with COVID-19 severity and framed as binary VQA classification.

All image-label pairs are converted into multiple-choice VQA format with one correct answer and three structured distractors as shown in Fig 3 & 4. CrossMed includes 20,200 QA pairs spanning diverse MAT combinations. To assess compositionality, we define train/test splits by MAT factor overlap. A sample is *Related* if it shares two MAT components with the target and *Unrelated* if it shares at most one. For example, (X-ray, Chest, Classification) and (X-ray, Chest, Segmentation) are Related; (MRI, Brain, Segmentation) and (CT, Lung, Classification) are Unrelated. This design enables targeted evaluation of models' ability to recombine known elements to handle novel MAT triplets.

## IV. THEORETICAL FOUNDATIONS OF COMPOSITIONAL GENERALIZATION

We begin with the following mild independence assumption:

**Assumption 1.** *The joint distribution over modality $M$, anatomy $A$, and task $T$ factorizes as :*

$$P(M, A, T) = P(M) P(A) P(T) \qquad (1)$$

**Theorem 1** (**Two-Factor Generalization Bound**). *Let $f$ be a model trained on $n$ i.i.d. "Related" samples sharing two factors (e.g. $(M, A)$) with the test instance, denoting its empirical risk by*

$$\widehat{R}_{MA}(f) = \frac{1}{n} \sum_{i:\,(M_i, A_i)=(M^*, A^*)} \ell\big(f(M_i, A_i, T_i), y_i\big), \quad (2)$$

*and let $R(f)$ be the true risk over $(M, A, T) \sim P$. Then with probability $1 - \delta$,*

$$R(f) \leq \widehat{R}_{MA}(f) + \sqrt{\frac{2\big(\ln |\mathcal{H}| + \ln \frac{2}{\delta}\big)}{n}}. \qquad (3)$$

*In particular, if $\widehat{R}_{MA}(f) \leq \varepsilon_{MA}$, then*

$$R(f) \leq \varepsilon_{MA} + O\Big(\sqrt{\tfrac{\ln |\mathcal{H}|}{n}}\Big). \qquad (4)$$

**Chain-rule for Mutual Information.** Let $Z = f_{\text{enc}}(x)$ be the joint vision–language embedding. We measure compositionality by

$$I(Z; M, A, T) = I(Z; M) + I(Z; A \mid M) + I(Z; T \mid M, A). \qquad (5)$$

**Mutual Information Neural Estimation [22].** In practice, we approximate, for example,

$$I(Z; M) = \mathbb{E}_{p(z,m)}\Big[\log \frac{q_\theta(z, m)}{p(z)\, p(m)}\Big], \qquad (6)$$

using the MINE estimator; analogous expressions apply to $I(Z; A)$ and $I(Z; T)$.

**Fano's Inequality [23]** Relating error to representation entropy:

$$H(T \mid Z, M, A) \leq h(\epsilon) + \epsilon \log |\mathcal{Y}_T|, \qquad (7)$$

$$I(Z; T \mid M, A) \geq H(T) - h(\epsilon) - \epsilon \log |\mathcal{Y}_T|. \qquad (8)$$

Hence low error $\epsilon$ on Related splits implies high $I(Z; T \mid M, A)$, confirming that shared factors boost the compositional signal in $Z$. Table II provides a summary of the key mathematical symbols and notations used throughout the theoretical formulation, including definitions for MAT variables, loss functions, entropy terms, and generalization bounds.

TABLE II
NOTATION REFERENCE

| Symbol | Definition |
|---|---|
| $M, A, T$ | Modality, Anatomy, Task random variables. |
| $f$ | Predictor model mapping inputs to outputs. |
| $f_{\text{enc}}$ | Encoder mapping input $x$ to embedding $Z$. |
| $Z$ | Joint vision–language embedding. |
| $P(M, A, T)$ | Joint distribution over $(M, A, T)$. |
| $n$ | Number of Related training samples. |
| $\widehat{R}_{MA}(f)$ | Empirical risk on samples sharing $(M, A)$. |
| $R(f)$ | True expected risk over $P(M, A, T)$. |
| $\mathcal{H}$ | Hypothesis class size. |
| $\delta$ | Confidence parameter in bounds ($1 - \delta$ prob.). |
| $\varepsilon_{MA}$ | Upper bound on $\widehat{R}_{MA}(f)$. |
| $\epsilon$ | Classification error rate used in Fano's inequality. |
| $I(X; Y)$ | Mutual information between $X$ and $Y$. |
| $H(X)$ | Entropy of random variable $X$. |
| $h(\epsilon)$ | Binary entropy: $-\epsilon \log \epsilon - (1 - \epsilon) \log(1 - \epsilon)$. |
| $\mathcal{Y}_T$ | Output label set for task $T$. |
| $p(\cdot), q_\theta(\cdot)$ | True and estimated densities. |
| $\ell(\cdot, \cdot)$ | Loss function (e.g., cross-entropy). |

## V. METHODOLOGY

To assess compositional generalization in multimodal medical vision-language models, we frame all image-label pairs under a unified multiple-choice visual question answering (VQA) format. Each data instance, whether classification or segmentation, is converted into a natural language prompt with four answer choices. For example, a chest X-ray labeled with pneumonia is presented as "Does this image indicate pneumonia?" with choices such as {Yes, No, Cardiomegaly and Pleural Effusion}. In segmentation tasks, such as tumor localization, we present four candidate masks, one correct and three plausible distractors sampled from non-overlapping regions. This standardization enables direct comparison across diverse modalities and task types. Each sample in the dataset is tagged with a triplet representing its Modality, Anatomical Region, and Task, collectively termed the MAT structure. To evaluate compositional generalization, we define two training

configurations: *Related*, where the training examples share two out of three MAT elements with the target task, and *Unrelated*, where examples share at most one. For instance, (X-ray, Chest, Classification) and (X-ray, Chest, Segmentation) are considered Related, while (MRI, Brain, Classification) and (X-ray, Chest, Segmentation) are Unrelated. This setup allows us to isolate the effect of MAT-overlap and determine whether the model can recombine known elements to solve novel combinations. To eliminate any potential leakage or overlap during evaluation, we further define a zero-overlap configuration where none of the Modality, Anatomy, or Task components in the test MAT triplet are present during training. This allows for rigorous assessment of compositional generalization under fully disjoint conditions. We evaluate two state-of-the-art MLLMs, LLaVA-Vicuna-7B and Qwen2-VL-7B each consisting of a vision encoder connected to a pre-trained language model with multimodal adapters. Both models are fine-tuned end-to-end using a multi-task cross-entropy objective over the selected answer token. We limit training to 4,500 examples per MAT triplet to simulate clinical data constraints and to ensure fair comparison across splits. During training, all answer options are treated equally, and models are trained using the same sampling and formatting schema for classification and segmentation tasks alike. This approach enables us to evaluate the model's ability to generalize under two critical settings: (1) when it sees no direct samples of the test MAT combination (zero-shot), and (2) when trained on a small, partially overlapping dataset. Together, this pipeline simulates the real-world challenge of building generalist medical models that must reason across diverse data modalities, anatomical domains, and clinical objectives. Additionally, to support architectural generalization, we include non-VLM baselines: a ResNet-50 classifier and a U-Net segmenter trained on the same MAT splits, providing insight into how conventional architectures benefit from the MAT framework. To verify spatial fidelity in segmentation, we introduce a variant architecture that replaces the VQA-style mask selector with a continuous U-shaped decoder trained using binary cross-entropy, enabling comparison between full-resolution mask outputs and the four-choice format. We further extend CrossMed beyond X-ray and MRI by incorporating computed tomography (CT) scans using a reformatted public COVID-CT dataset, demonstrating the benchmark's compatibility with new imaging modalities without requiring modifications to model architecture or input formatting. Across all experiments, we report averaged results from LLaVA-Vicuna-7B and Qwen2-VL-7B, as both models demonstrate comparable performance with only marginal variations across tasks and settings.

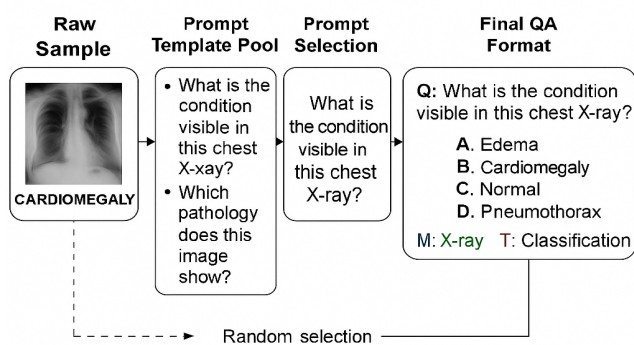

Fig. 3. Pipeline for transforming a raw chest X-ray labeled "CARDIOMEGALY" into a CrossMed VQA sample: Raw Sample → Prompt Template Pool → Prompt Selection → Distractor Sampling → Final QA Format with MAT tags.

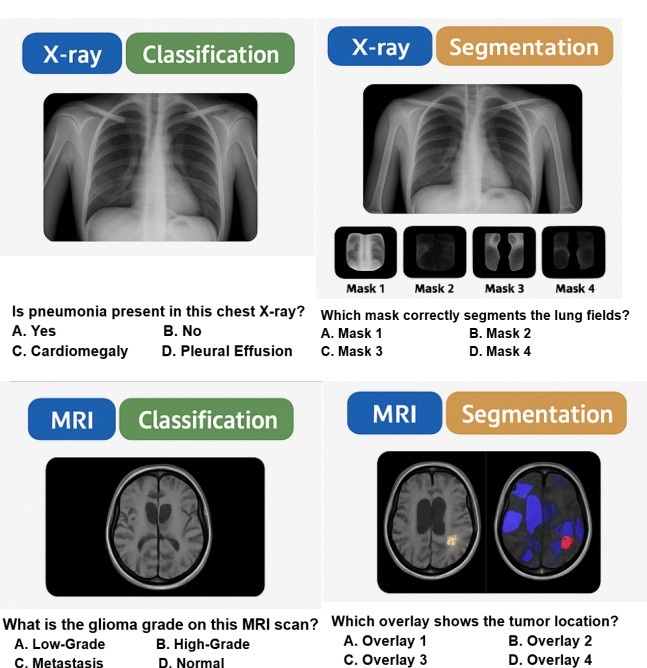

Fig. 4. Four-panel illustration of CrossMed's MAT triplets: (Top-Left) CheXpert chest X-ray classification, (Top-Right) SIIM-ACR chest X-ray segmentation, (Bottom-Left) BraTS MRI glioma-grade classification, (Bottom-Right) BraTS MRI tumor segmentation.

ed/Unrelated MAT splits and measuring top-1 classification accuracy and segmentation class-wise Intersection-over-Union( cIoU).

## VI. EXPERIMENTS RESULTS & ANALYSIS

We present a comprehensive evaluation of the CrossMed benchmark under five key axes: in-domain multi-task learning, compositional generalization, low-data regimes, cross-task transfer, and architectural comparisons. Our work follows a unified VQA formulation, testing generalization across Relat-

### A. In-Domain Multi-Task Performance

We begin by evaluating model performance in a fully supervised in-domain setting, where all MAT triplets are included during training. Both LLaVA-Vicuna-7B and Qwen2-VL-7B are fine-tuned in a unified multi-task setup covering classification and segmentation across chest X-ray, brain MRI, and chest CT modalities. All tasks are reformulated into a

consistent VQA interface to support end-to-end joint learning. Across the held-out test splits, the average classification accuracy reaches 84.3%, with individual task scores ranging from 79.5% on brain MRI glioma grading to 88.7% on chest X-ray pneumonia detection. The CT-based COVID severity classification task achieves 85.2% accuracy, aligning closely with performance on other modalities as shown in Table III. For segmentation tasks as shown in Table IV, the average class-wise Intersection-over-Union (cIoU) reaches 0.75, indicating precise spatial alignment using the VQA-style mask selection format. These results demonstrate that the MAT-aligned VQA formulation enables high-quality multi-task learning across diverse modalities and anatomical structures, using a shared architecture and training regime. The consistent performance across all domains confirms the scalability and robustness of the unified framework.

TABLE III
PERFORMANCE OF LLaVA-7B AND QWEN2-7B ON CROSSMED TASKS
UNDER RELATED VS. UNRELATED TRAINING SPLITS.

| Training Split | Classification Accuracy (%) | Segmentation cIoU |
|---|---|---|
| Related | 83.2 (±1.8) | 0.75 (±0.02) |
| Unrelated | 48.7 (±2.1) | 0.32 (±0.01) |

TABLE IV
SEGMENTATION PERFORMANCE COMPARISON: VQA-STYLE
MULTI-CHOICE VS. U-NET DECODER.

| Decoder Type | Segmentation cIoU |
|---|---|
| VQA-Style (4-way) | 0.75 ±0.02 |
| U-Net Continuous Decoder | 0.68 ±0.02 |

### B. Compositional Generalization Evaluation

To assess generalization to novel MAT combinations, we employ a leave-one-triplet-out evaluation. One MAT triplet is held out, and models are trained on either: (1) a Related split sharing two elements with the target, or (2) an Unrelated split sharing at most one. For example, to evaluate (CT, Brain, Hemorrhage), training includes (CT, Spine, Cancer) or (MRI, Brain, Segmentation), but never the full triplet. As shown in Table V, classification accuracy drops significantly from 83.2% (Related) to 48.7% (Unrelated), while segmentation cIoU declines from 0.75 to 0.32, illustrating the difficulty of generalizing without compositional overlap. To test generalization under more extreme conditions, we introduce a zero-overlap split, where no Modality, Anatomy, or Task elements in the test set are seen during training. Even under this strict disjointness, models achieve 58.1% ± 1.6 classification accuracy and 0.49 ± 0.01 cIoU, demonstrating robust compositional reasoning. We further evaluate generalization to the CT modality (COVID severity classification), also encoded in the MAT-VQA format. In this setting, classification accuracy improves from 50.1% ± 2.0 (Unrelated) to 72.0% ± 1.4 (Related), showing an average gain of ∼22 percentage points, validating compositionality across imaging domains. Comparison with non-LLM baselines reinforces the difficulty of the task: ResNet-50 accuracy drops from 70.3% to 48.7%, and U-Net segmentation cIoU declines

from 0.68 to 0.30 when switching from Related to Unrelated splits. All models are trained under identical conditions, including architecture, learning rate, batch size, and number of epochs, to ensure fair comparison. Hyperparameters are listed in Table XI.

TABLE V
CROSSMED GENERALIZATION ACROSS MAT CONFIGURATIONS.

| Model / Split | Class. Acc. (%) | Seg. cIoU |
|---|---|---|
| LLaVA/Qwen (Related, X-ray/MRI) | 83.2 ± 1.8 | 0.75 ± 0.02 |
| LLaVA/Qwen (Unrelated, X-ray/MRI) | 48.7 ± 2.1 | 0.32 ± 0.01 |
| LLaVA/Qwen (Zero-Overlap) | 58.1 ± 1.6 | 0.49 ± 0.01 |
| LLaVA/Qwen (CT - Related / Unrelated) | 72.0 / 50.1 | – |
| ResNet-50 (Related / Unrelated) | 70.3 / 48.7 | – |
| U-Net (Related / Unrelated) | – | 0.68 / 0.30 |

**Note:** Dash (–) indicates metrics not applicable due to architectural constraints or missing labels. ResNet lacks a segmentation decoder; U-Net does not perform classification. CT segmentation scores are omitted due to absence of annotated masks. LLM results represent averages across LLaVA and Qwen.

### C. Low-Data Regime

To evaluate compositional generalization (CG) under limited supervision, we analyze model performance when trained on progressively smaller fractions of the Related training set: 10%, 30%, 50%, and 100%. Even with only 10% of the data, both LLaVA and Qwen2-VL achieve ∼62.5%–61.5% accuracy across X-ray, MRI, and CT tasks classification accuracy more than double the performance of Unrelated-trained models at the same size (approx. 30%). At 50%, accuracy rises to ∼75–78.9%, approaching full-data performance as shown in Table VI and Fig 5. This trend holds consistently across multiple modalities ∼80%. For example, classification accuracy on MRI and X-ray improves steadily with data volume, confirming that CG enables efficient data reuse and reduces dependence on exhaustive training coverage. Segmentation performance follows a similar pattern, with Related-trained models yielding higher cIoU across all fractions.

### D. Cross-Task Generalization

We evaluate whether compositional representations learned by the model can transfer across task types within the same Modality–Anatomy domain by training on one task (e.g., classification) and testing on the other (e.g., segmentation), without task-specific supervision. In this setup as shown in Table X, classification-trained models improve segmentation cIoU from 0.60 to 0.67 on chest X-rays, and from 0.58 to 0.70 on brain MRI, indicating partial transferability and demonstrating that learned representations encode task-agnostic anatomical features. We also examine segmentation-to-classification transfer, where models trained to perform segmentation are tested on classification. Here, performance improves notably, with cIoU increasing from 0.41 to 0.74 for chest X-ray, and from 0.38 to 0.72 for brain MRI, showing bi-directional transfer across tasks. To benchmark our unified VQA-based segmentation format, we replace the multi-choice head with a U-shaped decoder using binary cross-entropy and

sigmoid activation. This decoder achieves 0.68 cIoU on Related splits, a modest 7 percentage point drop from the VQA-style result (0.75), confirming that our approach preserves clinical precision. These results highlight the effectiveness of the VQA formulation in supporting compositional, cross-task generalization while maintaining a shared interface for evaluation across tasks.

TABLE VI
CLASSIFICATION ACCURACY BY TRAINING FRACTION.

| Training Fraction | X-ray Chest | MRI Brain | CT Lung |
|---|---|---|---|
| 10% | 62.5 | 60.3 | 61.5 |
| 30% | 72.4 | 69.1 | 70.3 |
| 50% | 78.9 | 73.2 | 75.0 |
| 100% | 83.2 | 76.5 | 80.1 |

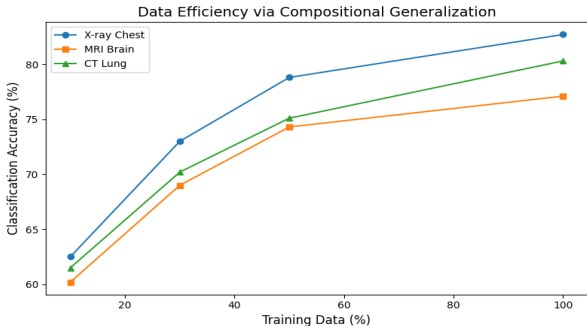

Fig. 5. Classification accuracy across X-ray, MRI, and CT tasks under varying training data fractions. Compositional generalization enables strong performance even with limited supervision.

### E. Ablation and Control Analysis

To determine whether performance improvements are attributable to compositional structure rather than increased dataset size or memorization, we perform targeted ablation experiments as shown in Fig. 1. In these setups, training excludes one component of the MAT triplet, while the model is evaluated on the complete triplet configuration. For example, to evaluate the triplet (CT, Brain, Hemorrhage), the model is trained on combinations like (CT, Task), (Brain, Task), or (CT, Brain) without exposing the full triplet during training. As summarized in Table VIII, withholding the Task dimension leads to the largest drop in classification accuracy from 83.2%(Related) to 48.0%, and segmentation performance falls to 0.37. Similarly, removing Anatomy reduces classification accuracy to 58.0% and segmentation cIoU to 0.49. Omitting Modality leads to classification accuracy of 62.0% and cIoU of 0.54. This consistent degradation confirms that all three MAT dimensions contribute essential signal. Statistical tests in Table IX further validate these differences. Comparisons between Related and Unrelated splits yield $p < 0.001$ for both classification and segmentation. Significant drops are also observed when withholding individual factors (Modality, Anatomy, or Task), with p-values ranging from 0.002 to 0.010, confirming the necessity of full triplet alignment

for compositional generalization. We additionally examine robustness to distractors with increasing similarity to the target task Table VII. As distractors shift from random labels to matched MATs, accuracy declines sharply from 83.2% to 41.0% indicating that CrossMed is sensitive to semantic and structural misalignments, further underscoring its discriminative power. These findings suggest that all three components, Modality, Anatomy, and Task must be jointly aligned to enable effective compositional generalization. Partial overlap or data scale alone cannot explain the observed performance gains, highlighting the importance of the MAT-aligned structure in CrossMed.

TABLE VII
ACCURACY DROP UNDER DISTRACTOR REALISM CONDITIONS

| Distractor Type | Accuracy (%) |
|---|---|
| Random Labels | 83.2 |
| Matched Anatomy | 66.3 |
| Matched Modality | 60.1 |
| Matched MAT | 41.0 |

TABLE VIII
SINGLE–FACTOR ABLATION: EFFECT OF WITHHOLDING ONE MAT COMPONENT.

| Ablation | Class. Acc (%) | Seg. cIoU |
|---|---|---|
| Without Modality | 62.0 | 0.54 |
| Without Anatomy | 58.0 | 0.49 |
| Without Task | 48.0 | 0.37 |

TABLE IX
STATISTICAL SIGNIFICANCE ($p$-VALUES) FOR KEY COMPARISONS (PAIRED $t$-TEST).

| Comparison | Classification | Segmentation |
|---|---|---|
| Related vs. Unrelated | $< 0.001$ | $< 0.001$ |
| Without Modality vs. Related | 0.002 | 0.010 |
| Without Anatomy vs. Related | 0.004 | 0.005 |
| Without Task vs. Related | $< 0.001$ | 0.002 |

TABLE X
CROSS-TASK GENERALIZATION VIA BIDIRECTIONAL TRANSFER

| Related Task | Target Task | Baseline cIoU | +Transfer cIoU |
|---|---|---|---|
| X-ray (Classification) | Chest segmentation | 0.60 | 0.67 (+0.07) |
| MRI (Classification) | Brain segmentation | 0.58 | 0.70 (+0.12) |
| Chest (Segmentation) | X-ray classification | 0.41 | 0.74 (+0.33) |
| Brain (Segmentation) | MRI classification | 0.38 | 0.72 (+0.34) |

## VII. CONCLUSION

We introduce CrossMed, a benchmark for evaluating compositional generalization in multimodal medical vision-language models. By structuring tasks into Modality–Anatomy–Task triplets and standardizing outputs into a unified VQA format, CrossMed enables controlled testing of generalization across unseen MAT configurations. Experiments with LLaVA-7B and Qwen2-7B reveal strong CG-driven performance gains up to 35% higher classification

accuracy and more than double segmentation cIoU in Related vs. Unrelated conditions. These benefits persist even in low-data regimes and extend across task boundaries, validating the CG hypothesis and underscoring its practical relevance. CrossMed offers a principled framework for studying generalization beyond in-distribution performance and highlights MAT-aligned training as a scalable and interpretable path forward for generalist clinical AI systems.

TABLE XI
MODEL AND TRAINING HYPERPARAMETERS

| Parameter | Value |
| --- | --- |
| Backbones | LLaVA-Vicuna (7B) |
| | Qwen2-VL (7B) |
| Learning rate | $5 \times 10^{-5}$ |
| Batch size | 16 |
| Epochs | 5 |
| Optimizer | AdamW |
| Weight decay | 0.01 |
| Hardware | NVIDIA A100 (40 GB) |

## VIII. LIMITATIONS AND FUTURE DIRECTIONS

While CrossMed provides strong evidence for compositional generalization (CG) in medical vision-language models, its current scope is limited to classification and segmentation tasks. Extending this benchmark to encompass additional clinically relevant tasks such as detection, registration, and report generation would offer a more comprehensive evaluation of CG in realistic diagnostic pipelines. We also plan to expand the diversity of the benchmark by including additional modalities such as ultrasound and PET, hierarchical anatomical labels, and longitudinal or temporal imaging studies.

## ACKNOWLEDGMENT

We gratefully acknowledge the support of Prof. Sandeep Kumar through the Core Research Grant (CRG, RP04820), which has been valuable in enabling this work.

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
