# OpenReview forum: "CrossMed: A Multimodal Cross-Task Benchmark for Compositional Generalization in Medical Imaging"
_IEEE.org/EMBS/BHI/2025/Conference — BHI 2025_

### Official Review · Reviewer_xtfQ · 2025-07-16
**Review of paper 285**

**Confidence:** 3
**Clarity Of Writing:** great
**Clinical Significance:** fair
**Methodological Novelty:** fair
**Overall Rating:** 6
**Final Rating:** 7

**Experiments And Results:**

good

**Questions For The Authors:**

- Why have the authors converged on the task format presented here? Have questions where continuous functions are evaluated been considered?
- Besides the two VLM models used in this work, have the authors attempted to validate any other model?
- Do the authors plan to release the code used for training and running experiments upon publication?

**Strengths:**

- The proposed dataset will be useful in future studies evaluation VLM model generalizability.
- The paper is well written, has a solid experimental section and a theoretical justification for the considered setting

**Summary Of The Paper:**

The authors introduce a new benchmark dataset for medical vision language models by extending existing medical datasets under the modality-anatomy-task framework. Using this new dataset, the authors fine-tune existing VLM models in order to demonstrate that models can compose learned modality, anatomy, and task cues to solve entirely unseen MAT combinations. Additional experiments and brief theoretical analysis are provided in support of this idea.

**Weaknesses:**

- In practical settings, vqa models may already have the MAT triplets of interest in their training data. Thus, model generalizability to new combinations may be limited in utility.
- The task format may be too limiting for wider generalization - i.e. questions with only 4 answers
- Please fix the text scaling in figure 5

---

### Official Review · Reviewer_PHMY · 2025-07-17
**CrossMed: A Multimodal Cross-Task Benchmark for Compositional Generalization in Medical Imaging**

**Confidence:** 4
**Clarity Of Writing:** good
**Clinical Significance:** good
**Methodological Novelty:** good
**Overall Rating:** 6

**Experiments And Results:**

good

**Questions For The Authors:**

1. Can this benchmark support zero-shot generalization where a task (not just a MAT configuration) is entirely new?
2. How does CrossMed’s VQA framing for segmentation compare to standard segmentation metrics in terms of clinical relevance?
3. Would the observed gains from related training hold if distractor answers were less structured or less plausible?

**Strengths:**

The creation of a structured and scalable benchmark addressing generalization across unseen combinations of modality, anatomy, and task provides a valuable tool for the field. This aligns closely with the conference focus on processing and interpretation of multi-modal biomedical data. The experiments show clear benefits from compositional structure. Related MAT training improves performance significantly over Unrelated data. These results support the hypothesis that CG is a key inductive bias for MLLMs in healthcare. The finding that even 10% of related data can outperform full-scale unrelated training is valuable for real-world medical applications with limited labels. The use of two independent MLLMs and consistent performance across them strengthens the paper’s generalizability.

**Summary Of The Paper:**

The paper presents CrossMed, a new benchmark developed to evaluate compositional generalization (CG) in multimodal large language models (MLLMs) within medical imaging. CrossMed reformulates existing datasets (CheXpert, SIIM-ACR, and BraTS 2020) into a unified visual question answering (VQA) format, each labeled by a Modality–Anatomy–Task (MAT) triplet. The dataset comprises 19,200 instances across classification and segmentation tasks. Two MLLMs—LLaVA-Vicuna-7B and Qwen2-VL-7B—are evaluated across training conditions that vary in MAT similarity to test data (Related vs. Unrelated splits). The study measures performance in in-domain, compositional, low-data, and cross-task scenarios. It also introduces theoretical bounds on CG using mutual information and provides ablation studies to isolate the contributions of individual MAT components.

**Weaknesses:**

The benchmark currently focuses only on classification and segmentation. Detection, report generation, and longitudinal tasks are not covered, limiting the conclusions about general-purpose clinical AI applicability. Converting all tasks to VQA format may obscure domain-specific nuances. For instance, the clinical utility of selecting a correct segmentation mask from four distractors is different from pixel-wise evaluation used in practice. The segmentation task is reduced to multiple-choice selection rather than continuous spatial overlap measures beyond cIoU. This could limit interpretability in clinical use. While the study includes X-ray and MRI, it lacks CT, ultrasound, and temporal imaging, which are important in many diagnostic workflows. Although two strong models are used, the paper does not evaluate whether specific architectural choices contribute to CG, nor whether smaller or less capable models exhibit similar trends.

---

### Official Review · Reviewer_5sn9 · 2025-07-17
**Benchmark for Multimodal LLMs**

**Confidence:** 3
**Clarity Of Writing:** good
**Clinical Significance:** good
**Methodological Novelty:** great
**Overall Rating:** 7

**Experiments And Results:**

great

**Questions For The Authors:**

Thank you for this interesting paper on MLLM. The compositional generalization is an interesting topic, and I believe highly valuable for the rapidly evolving medical AI community. I am interested to know why unrelated split has low accuracy.

**Strengths:**

Certainly, this is a new topic to understand compositional generalization on multimodal LLMs. Model has good classification accuracy for related training split.
Model performed well even with 10% of related data.
Introduction and related work sections are written well.

**Summary Of The Paper:**

The paper discusses about CrossMed - a benchmark to evaluate cross training and Compositional Generalization in Medical Imaging. This paper appears interesting because it talks about cross training and generalizing the LLM model for various modalities. Paper uses 2 vision-language models to compare/evaluate and identifies that structuring tasks into MAT (modality-anatomy-tasks) triplets.

**Weaknesses:**

1) I was able to find very similar paper: https://arxiv.org/pdf/2412.20070
Exploring Compositional Generalization of Multimodal LLMs for Medical Imaging

2) Even though, it says generalization, as mentioned in the limitation, this paper talks only about classification and segmentation. How this model works for disease prognosis or treatment guidance.

---

### Official Review · Reviewer_qWDo · 2025-07-18
**The paper presents a valuable benchmark for fine-grained cross-modal reasoning in medical AI but is limited by its narrow task scope, lack of baseline comparisons, potential oversimplification through task reformulation, and omission of relevant prior work.**

**Confidence:** 2
**Clarity Of Writing:** good
**Clinical Significance:** good
**Methodological Novelty:** fair
**Overall Rating:** 3
**Final Rating:** 5

**Experiments And Results:**

good

**Questions For The Authors:**

Please see weaknesses

**Strengths:**

The paper introduces CrossMed, a benchmark specifically designed to evaluate and advance fine-grained cross-modal reasoning in the medical domain, filling a critical gap in multimodal medical AI research.

Its significance lies in systematically challenging foundation models with domain-specific, complex reasoning tasks involving image-text alignment, thereby pushing the boundaries of medical AI understanding and evaluation.

**Summary Of The Paper:**

This paper presents CrossMed, a controlled benchmark designed to isolate compositional generalisation (CG) signals in multimodal medical tasks. It curates 4 carefully selected public datasets, resulting in 19,200 high-quality QA pairs. By focusing on only 4 MAT (Modality-Anatomy-Task) triplets, CrossMed ensures a clean and interpretable setup to study CG in isolation. The benchmark covers classification and segmentation tasks, evaluated using LLaVA‑Vicuna‑7B and Qwen2‑VL‑7B vision-language backbones. Paper's key finding is that even this small but clean MAT overlap can yield substantial CG improvements, demonstrating the value of structured generalisation even in compact settings.

**Weaknesses:**

1. Even though the paper claims to provide a comprehensive dataset, it restricts itself to only two tasks: classification and segmentation, and a few two modalities (X-ray and MRI)

2. MLLMs demonstrate strong CG, with models trained on Related splits achieving 82.2% classification accuracy and 0.75 segmentation IoU =>> What’s the baseline performance of task-specific models? Is there non-MLLM architectures like CNNs or transformers trained on the same datasets, which is a critical oversight missing ?

3. Reformulating CheXpert, SHIN-ACR, and BRATS 2020 into a multiple-choice VQA format is a clever trick, but by forcing diverse tasks into a uniform framework, you risk oversimplifying the problem space and introducing artifacts. For instance, converting segmentation into a multiple-choice mask selection task minimizes the pixel-level complexity of segmentation, potentially inflating performance metrics like IoU. How does this reformulation might bias results or affect clinical relevance ?

I was a bit surprised that this paper does not cite the following work: "On the Compositional Generalization of Multimodal LLMs for Medical Imaging." I also found a few similarities in the concepts presented in this paper and the referenced paper.
https://arxiv.org/pdf/2412.20070v1